# Passive seismic imaging of the Lower Palaeozoic in the Sudret area of Gotland, Sweden

Zhihui Wang<sup>1,2</sup>, Christopher Juhlin<sup>2,\*</sup>, Peter Hedin<sup>3</sup>, Mikael Erlström<sup>4</sup>, Daniel Sopher<sup>3</sup>

<sup>1</sup>Chinese Academy of Geological Sciences, Beijing, 100037, China

<sup>2</sup>Uppsala University, Department of Earth Sciences, Uppsala, 75236, Sweden

<sup>3</sup>Geological Survey of Sweden, Uppsala Office, Uppsala, 75128, Sweden

<sup>4</sup>Lund University, Department of Geology, Lund, 22362, Sweden

Correspondence to: Christopher Juhlin (christopher.juhlin@geo.uu.se)

Abstract. Passive seismic data were acquired together with active seismic data along a 2.8 km long profile in the Sudret area of Gotland, Sweden, as part of a feasibility study for storage of CO<sub>2</sub> below the Baltic Sea. Seismic interferometry using cross-correlation and cross-coherence was employed on the passive seismic data. Cross-correlation was used to retrieve virtual shot gathers containing mainly surface waves, while cross-coherence was used to retrieve mainly seismic reflections. Inversion for shear wave velocity and CDP processing of the passive data result in velocity profiles and images that correlate well with borehole data, synthetic seismograms and active seismic data acquired at the site. Interpretations of the passive surface wave and body wave results provide geological information which complement the active data results, the surface waves providing S-wave velocity information and the body waves providing an image that may have better signal quality at deeper levels. Results from the active seismic and passive seismic data correlate well and there is no indication of any large-scale faults in the area. Furthermore, analysis of the frequency and direction of the ambient noise using power spectral density and beam forming show that ocean waves and human activity around the island of Gotland make the Sudret area an ideal location for passive imaging through ambient noise interferometry. Our results illustrate that passive seismic imaging can be an important complement to active seismic data for structural studies of the subsurface with respect to CO<sub>2</sub> storage and monitoring in the Gotland area, Sweden, and perhaps elsewhere.

### Keywords

Passive seismic; Carbon capture and storage; Ambient noise; Seismic interferometry; Virtual shot gathers

#### 25 1 Introduction

Carbon capture and storage (CCS) is a strategy that can be employed for reducing atmospheric emissions of greenhouse gases and thereby their adverse effects on the climate (Niemi et al., 2017). A large theoretical capacity to store carbon dioxide (CO<sub>2</sub>) in the Palaeozoic sedimentary successions within the Baltic Basin exists, including large saline aquifers and depleted oil and gas fields (Anthonsen, 2013; Sopher et al., 2014; Shogenova et al., 2021). Identification and characterization of potential storage sites are important cornerstones for establishing safe geological storage of CO<sub>2</sub> (Lüth et al., 2017). In

删除[王志辉 [2]]: C

删除[evahj56@hotmail.com]: the

删除[evahj56@hotmail.com]: Both the

删除[evahj56@hotmail.com]: i

删除[Chris Juhlin]: more

删除[evahj56@hotmail.com]: deeperlower frequency

删除[王志辉]: have

删除[Chris Juhlin]: obtained by analysing

删除[Chris Juhlin]: data

删除[Chris Juhlin]: are in a good

删除[Chris Juhlin]: ion

删除[王志辉]: The passive data are consistent with the active data

删除[王志辉]: s

删除[王志辉]: s

删除[王志辉]: passive seismic

删除[王志辉]: evaluating the

删除[evahj56@hotmail.com]:,

Sweden, sedimentary strata potentially suitable for CO<sub>2</sub> storage are only found in the Baltic Basin, south of Gotland and in southwest Skåne and adjacent offshore areas. During the past few years, the Geological Survey of Sweden (SGU) has been investigating suitable locations for CO<sub>2</sub> storage offshore. Two Swedish marine areas with potential for storage are being investigated, one south of Skåne in southern Sweden and one in the southeastern part of the Baltic sea (www.sgu.se). These areas contain deeply seated porous reservoir sandstones capped by thick caprock which could provide the prerequisites for safe storage of CO<sub>2</sub>. Because drilling at sea is costly, as well as complex, two boreholes were drilled down to about 800 m in the Sudret area on south Gotland where the sandstones are found deepest onshore and can be used as analogs to the offshore area. In November, 2023, reflection seismic data were acquired in the vicinity of the two boreholes to provide a better understanding of the sedimentary strata and the local subsurface structural framework (Juhlin et al., 2025).

40

Active seismic reflection surveying and distributed acoustic sensing (DAS) with high resolution have been the dominant methods for imaging and monitoring CO<sub>2</sub> storage sites (Juhlin et al., 2007; Alcalde et al., 2014; Pevzner et al., 2015; Roach et al., 2015; White et al., 2015, 2022; Huang et al., 2016; Zhang et al., 2016; Cheraghi et al., 2017; Harris et al., 2017; Lüth et al., 2017; Ivandic et al., 2018; Papadopoulou et al., 2023, 2024; Wang and Lawton, 2024; Zappalà et al., 2024). However, active-source seismic surveys suffer from the high cost of data acquisition and lack of continuous monitoring for sudden temporal variations (Ikeda et al., 2017). Several studies show that passive seismic surveying could be a complement to CO<sub>2</sub> geological storage monitoring (De Ridder and Biondi, 2012; Riahi et al., 2013; Boullenger et al., 2015; Gassenmeier et al., 2015; Cheraghi et al., 2017; Cao and Askari, 2019; Xu et al, 2012; Hassing et al., 2024). Potential advantages of the technique are that it is relatively environmentally, friendly and cost effective, as it uses ambient noise as a source.

50

Our investigation recorded 14-hours of continuous passive data along a 2.8 km long profile to retrieve virtual shot gathers using seismic interferometry. First we present examples of ambient noise data that include surface waves and body waves. These data are analysed for their frequency content and directionality to determine if seismic interferometry is suitable to apply to the data set. Secondly, cross-correlation and cross-coherence calculations were performed to retrieve surface waves and body waves, respectively. Subsequently, standard surface wave and seismic reflection data processing were employed to obtain a shear wave velocity model and a seismic reflection stacked section. Both the body waves and surface waves provide high-quality images and have good consistency with the borehole section, geophysical logging and the active seismic data from the same location. Moreover, the results reveal some potential deep geological information which the active data could not provide due to energy limitations of the 500 kg weight drop hammer used as a source. Finally, we discuss the comparisons of active and passive data results and the relevance of our results for CO2 storage and seismic imaging.

```
删除[Chris Juhlin]:,
```

删除[evahj56@hotmail.com]:

删除[evahj56@hotmail.com]: ed

删除[evahj56@hotmail.com]: d

删除[evahj56@hotmail.com]: and

删除[evahj56@hotmail.com]: the data

删除[evahj56@hotmail.com]: applied

删除[Chris Juhlin]: reliable

删除[Chris Juhlin]: because of

删除[evahj56@hotmail.com]: the

删除[王志辉]: the

删除[evahj56@hotmail.com]: a

删除[王志辉]: limited

删除[王志辉]: energy

删除[Chris Juhlin]: show that

设置格式[王志辉]:字体:(中文)黑体,(中文)中文(简体)

设置格式[王志辉]:字体:(中文)黑体,(中文)中文(简体)

删除[evahj56@hotmail.com]: 2

设置格式[王志辉]:字体:(中文)黑体,(中文)中文(简体)

### **2** Geological setting of the test site

The test site is situated on Sudret, the southernmost part of Gotland Island, which lies in the central Baltic Sea (Fig. 1a). The present investigation is furthermore focused around the cored borehole Nore-1 drilled in 2024 to 791 m depth (Erlström et al., 2024). Nore-1 and the neighboring Nore-2 borehole were drilled by the Geological Survey of Sweden in 2023 within a CO<sub>2</sub> investigation program launched by the Swedish Government with the objective to investigate three potential Cambrian sandstone reservoirs, as well as studying the overlying Ordovician and Silurian sealing strata. The Nore-1 borehole intersected a 778 m thick sedimentary succession on top of porphyritic potassium-rich granite basement (Fig. 1b and 1c). At the base of the sedimentary succession there is a c. 220 m thick Lower and Middle Cambrian interval which includes three 20 to 50 m thick sandstone units, i.e. the Viklau, När and Faludden sandstones, with intermediate layers of silty claystone and mudstone. The three potential reservoirs for CO<sub>2</sub> storage are located at the depths of 554,575 m, 677,706 m and 728,778 m, respectively. The uppermost unit, i.e. the Faludden Sandstone, is the most promising candidate for CO<sub>2</sub> storage. The overlying Ordovician interval between 472 m and 554 m depth comprises variably argillaceous limestone and calcareous mudstone. The top of the Ordovician has strong reflectivity according to results of the active seismic data analysis (Juhlin et al., 2025). One interval in the Ordovician, between 492 and 505 m depth, stands out due to its bentonite and mudstone layers. This unit has been consistently penetrated in existing wells on Gotland and can be identified in most of the available geophysical well logs. Furthermore, with relatively high frequency seismic reflection data, a reflection can be observed to be associated with this interval (Erlström and Sopher, 2019). Overlying the Ordovician is a 472 m thick Silurian succession, which from 118 m depth down to the Ordovician mainly comprises marlstone and calcareous claystone with subtle lithological variations. Besides a 55 m thick mudstone above the Ordovician, and two relatively thin limestone-dominated intervals at 400 m and 310 m, which can be correlated between wells, most of the interval between 118-472 m is poorly lithostratigraphically defined. The uppermost part of the Silurian from 118 m depth to the top of the bedrock consists of a mixed interval with limestone, mudstone, sandstone and reef limestone, representing the Eke, Burgsvik and Hamra-Sundre formations. The Eke limestone between 107 m and 118 m stands out as a significant marker in the geophysical logs in all wells on south Gotland. Above the Eke limestone follows the c. 50 m thick Burgsvik interval with alternating layers of finegrained sandstone and calcareous mudstone. The uppermost part of the Silurian present in the Sudret area consists of relatively coarse-grained carbonates of the Hamra and Sundre formations. The Quaternary deposits are up to c. 4 m thick in the study area and mainly consist of sandy and pebbly deposits.

删除[王志辉]: why the frequency content and direction of the ambient noise makes the Sudret area an ideal location for passive imaging.

删除[王志辉]: The test site is located on Sudret, the southernmost part of the island of Gotland, located in the central part of the Baltic Sea

删除[evahj56@hotmail.com]: the

删除[evahj56@hotmail.com]: the

删除[王志辉]: -

删除[王志辉]:

删除[王志辉]: -

删除[王志辉]: The top of the Ordovician generates a clear reflection in the active seismic data.

删除[Chris Juhlin]: is

删除[Chris Juhlin]: between

删除[Chris Juhlin]: s

删除[王志辉]: This unit is consistent between wells on Gotland and can be identified in most of the available geophysical well logs. Furthermore, with relatively high frequency seismic reflection data, a reflection can be observed to be associated with this interval (Erlström and Sopher, 2019).

删除[王志辉]: -

Figure 1: (a) The experimental site shown as the red star located in southern Gotland, Sweden. (b) Locations of survey line and borehole Nore-1. For clarity, every tenth receiver and source location are shown. Receivers are marked by the yellow triangles, and source locations are represented by the orange triangles. Both receiver spacing and source spacing were 10 m. The blue hollow circle shows the Nore-1 borehole location (the aerial image from Google Maps). (c) Mapped Nore-1 borehole lithology.

#### 3 Data acquisition and processing

## 3.1 Active and passive data acquisition

In order to image the subsurface down to the Precambrian basement in the vicinity of the two boreholes that had been drilled earlier and to investigate if any faults could be observed in the sedimentary strata a 2D active seismic survey was carried out from 12 November, 2023 to 13 November, 2023. This 2D survey was part of a larger investigation of the area that included a small 3D survey (Juhlin et al., 2025). A skid-steer loader with a 500 kg weight drop hammer with iron plate was used as a source and 329 5 Hz SmartSolo nodal units were available for recording. After acquiring the active data, ambient noise data from 17:00 on 12 November, 2023 to 07:00 on 13 November, 2023 (UTC+1) were recorded along the 2.8 km long profile with 10 m receiver spacing and 1 ms sample rate (Table 1).

删除[王志辉]: 14-hours of continuous passive data

删除[王志辉]: (Table 1)

删除[王志辉]:

删除[王志辉]: Wang et al., 2024

Table 1: Active and passive seismic data acquisition parameters for the Sudret area on Gotland, Sweden

|                         | Passive Seismic Data                              | Active Seismic Data   |  |
|-------------------------|---------------------------------------------------|-----------------------|--|
| Recording system        | Smart Solo Smart Solo                             |                       |  |
| Receiver                | IGU-16HR 1C, 5Hz                                  | IGU-16HR 1C, 5Hz      |  |
| Source                  | 500 kg weight drop hammer                         |                       |  |
| Receiver interval       | 10 m                                              | 10 m                  |  |
| Shot interval           | 10 m (virtual shot)                               | 10 m                  |  |
| CMP interval            | 5 m                                               | 5 m                   |  |
| Sampling rate           | 1 ms                                              | 1 ms                  |  |
| Recording length        | 14 hours                                          | 2 s                   |  |
| Minimum offset          | 0 m                                               | 0 m                   |  |
| Maximum offset          | 2 <mark>79</mark> 0 m                             | 2 <mark>79</mark> 0 m |  |
| Survey geometry         | Asymmetric split spread, fixed-geophone locations |                       |  |
| Number of receivers 280 |                                                   | 280                   |  |
| Number of source points | 280 (virtual shot)                                | 277                   |  |

### 3.2 Ambient noise data analysis

Two continuous data recordings were picked out to demonstrate surface waves and body waves included in the ambient noise data (Fig. 2). Surface waves are clear and have a velocity of about 1500 m/s in Figure 2a. Body waves are dominant after filtering out low frequency signals with a bandpass filter of 5-6-20-25 Hz and their linear apparent velocity is about 4000 m/s (Fig. 2b). To better understand the time-variant nature of the ambient noise sources, 14 one-hour power spectral density panels (Fig. 3) were calculated for every two stations from receiver No. 79 to No. 358 from 17:00 on 12 November, 2023 to 07:00 on 13 November, 2023 (UTC+1). All plots show good consistency except the last one at 06:00-07:00, indicating the frequency components of the noise sources were stable during data acquisition. Unlike the active seismic data with certain energy and location of sources, passive data relies on natural and anthropogenic sources, such as seismic energy generated from ocean-wave-induced microseisms (0.03-1 Hz), distant earthquakes (< 1 Hz), traffic (> 1 Hz) and so on (Nimiya et al., 2021; Bertoldi et al., 2024). In our study, ambient noise data were observed at frequencies of 0.5-20 Hz in each panel. Four peaks at the frequencies of 1 Hz, 5 Hz, 12 Hz and 16 Hz originate from ocean waves and human activities. These frequencies all contribute to retrieving the virtual source gathers. Surface waves dominate at the frequencies of 0.5-7 Hz, whereas body waves dominate at 7-20 Hz.

删除[王志辉]: (after Wang et al., 2024)

删除[王志辉]: 80

删除[王志辉]: 80

删除[王志辉]: 2

删除[Chris Juhlin]:

删除[王志辉]:

删除[王志辉]: -

删除[王志辉]: -

删除[王志辉]:

删除[王志辉]:

删除[王志辉]:

删除[王志辉]:

删除[王志辉]: Two continuous data recordings were picked out to demonstrate surface waves and body waves included in the ambient noise data after applying different bandpass filters (Fig. 3). Surface waves are clear and have a velocity of about 1500 m/s in Fig. 3a. Body waves are dominant after filtering out low frequency signals with a bandpass filter of 5-6-20-25 Hz and their linear apparent velocity is about 4000 m/s (Fig. 3b).

Figure 2: Continuous ambient noise data recordings at different times to display surface waves and body waves. (a) Ambient data recording dominated by surface waves after applying a bandpass filter of 0.5-1-40-50 Hz. (b) Ambient data recording dominated body waves after applying a bandpass filter of 5-6-20-25 Hz.

Figure 3: Amplitude spectra normalization of one-hour ambient noise recordings continuously from 17:00 on 12 November, 2023 to 07:00 on 13 November, 2023 (UTC+1).

After the power spectrum density analysis, we evaluate the spatial distributions of the ambient noise sources at the experimental site. A beam forming analysis (Gouédard et al., 2008; Cheraghi et al., 2015) of receiver stations No. 6, No. 49 and No. 185, forming a triangle, was performed with 14-hours of continuous passive data. Fig. 4a and Fig. 4b represent the maximum beam power in the frequency ranges of 0.5-7 Hz and 7-20 Hz, respectively. The recorded wavefield of surface waves with a velocity less than 4000 m/s came from the NNW and the azimuth of the ambient noise sources are consistent with the seismic survey line (green lines) and located in the stationary-phase regions. This implies the Green's function and its time-reversed variant would be retrieved equally well. The ambient noise sources for the body waves are scattered in several directions at a velocity higher than 3000 m/s and related to anthropogenic sources originating from human activities around the island of Gotland, such as in mainland Sweden, Finland, Estonia, Poland, Germany, Denmark, Norway, etc (Fig. 1a). This implies that body waves probably can be retrieved from our ambient noise data.

Figure 4: Directional beam forming analysis of ambient sources with different frequency components. (a) The maximum values of normalized strength of source energy at the frequencies of 0.5-7 Hz. (b) The maximum values of normalized strength of source energy at the frequencies of 7-20 Hz. The green lines show the azimuth of the seismic profile.

## 33 Data pre-processing

Prior to applying seismic interferometry, standard data pre-processing was applied to the ambient noise data (Table 2). We removed the mean and de-trended the recordings and afterwards separated 14 hours of continuous ambient noise data into 8400 segments with 6 s length. Then, normalization and bandpass filtering with different parameters were applied to extract the surface and body waves from the ambient noise data. For the surface waves, a bandpass filter with corner frequencies of 0.5-1-40-50 Hz was used to filter out too low/high-frequency components to avoid producing noise during the cross-correlation calculation. Prior to cross-correlation, normalization in the time domain was applied. For the body waves, a bandpass filter with corner frequencies of 5-6-20-25 Hz and spectral whitening in the frequency domain were applied to suppress surface waves and enhance reflected energy.

删除[Chris Juhlin]: at

删除[Chris Juhlin]: located from

删除[王志辉]: -7

删除[王志辉]: 7

删除[王志辉]: -

删除[王志辉]:

设置格式[王志辉]:字体:(中文)宋体,英语(美国)

删除[王志辉]: 2

设置格式[王志辉]:字体:(中文)宋体

设置格式[王志辉]:字体:(中文)宋体,英语(美国),(

删除[evahj56@hotmail.com]: separate

设置格式[王志辉]:字体:(中文)黑体,英语(美国),([

设置格式[王志辉]:字体:(中文)黑体,英语(美国),(...

设置格式[王志辉]:字体:(中文)黑体,英语(美国)。(「

设置格式[王志辉]:字体:(中文)黑体,英语(美国),(

删除[evahj56@hotmail.com]: and followed

删除[evahj56@hotmail.com]: by

删除[evahj56@hotmail.com]: before cross-correlation

Table 2. Passive data processing workflow and parameters.

| - C. | Processing workflow                |                                                    |              |
|------|------------------------------------|----------------------------------------------------|--------------|
| Step | Surface waves                      | <b>Body waves</b>                                  | 带格式表格[王志辉]   |
| 1    | Data input, read SAC format data   | Data input, read SAC format data                   | 删除[王志辉]: EGD |
| 2    | Data segment (6 s)                 | Data segment (6s)                                  | 删除[王志辉]: EGD |
| 3    | De-meaning and De-trending         | De-meaning and De-trending                         | '            |
| 4    | Band pass filter (0.5-1-40-50 Hz)  | Band pass filter (5-6-20-25 Hz)                    |              |
| 5    | Time domain normalization, one-bit | Frequency domain normalization, spectral whitening |              |
| 6    | Cross <sub>3</sub> correlation     | Cross-coherence                                    | 删除[王志辉]:     |
| 7    | Dispersion curve calculation       | Geometry, CMP binning: 5m                          | 删除[王志辉]:     |
| 8    | Velocity inversion                 | Spectral Equalization, 5-6-18-19Hz                 |              |
| 9    | Data output                        | Median filter, airwaves 340 m/s,                   |              |
| 10   |                                    | Surgical muting, mute above first breaks           |              |
| 11   |                                    | Velocity analysis                                  |              |
| 12   |                                    | NMO, 70% stretch                                   |              |
| 13   |                                    | Stack                                              |              |
| 14   |                                    | F-X deconvolution, window length 19 traces         |              |
| 15   |                                    | Band-pass filter 5-6-18-19 Hz                      |              |

# 155 3.4 Retrieving Green's function by cross-correlation and cross-coherence

The wavefield generated by a noise source can be represented by the convolution of the source wavelet and a Green's function. Since Claerbout (1968) first proposed that the reflection response of a horizontally layered medium can be obtained from one side of the autocorrelation of its transmission response, different seismic interferometry techniques have been developed and improved to retrieve the Green's function and characterize the seismic wave propagation between two receivers (Cole, 1995; Rickett and Claerbout, 1999; Schuster, 2001; Campillo and Paul, 2003; Bakulin and Calvert, 2006;

删除[王志辉]: 3

删除[王志辉]: '

Draganov et al., 2009; Wapenaar et al., 2004, Shapiro et al., 2005; Snieder, 2004; Wapenaar and Fokkema, 2006; Prieto et al., 2009; Nakata et al., 2015; Meles et al., 2015; Olivier et al., 2015; Oren and Nowack, 2017). However, the different methods have both strengths and weaknesses, for example, cross-correlation is stable, clearer and less noisy and cross-coherence can enhance temporal resolution and be able to retrieve body wave data (Snieder et al., 2009, Nakata et al., 2011, Zhang et al., 2019). Here, we use cross-correlation (Wapenaar and Fokkema, 2006) and cross-coherence (Nakata et al., 2011) of the ambient noise data recorded at different receivers to retrieve approximations of Green's functions (Fig. 5) after data preprocessing. Strong surface waves are observed on the causal and acausal parts and airwaves are only retrieved on the acausal part through the cross-correlation calculation (Fig 5a). The symmetry between the causal and acausal parts of the crosscorrelation functions suggests a noise directionality that correlates well with the directional beam forming analysis (Fig 4a). The body wave signatures of cross-coherence functions are shown in Figure 5b. The direct or diving S-wave (SPW) and airwaves are clear in the acausal part of positive-offset traces and causal part of negative-offset traces (quadrant 1 and guadrant 3 in Fig. 5b), while the direct or diving P-wave (DPW) and reflections are observed in the causal part of positiveoffset traces and acausal part of negative-offset traces (quadrant 2 and quadrant 4 in Fig. 5b). Existing research has demonstrated that more complete results are obtained when the causal and acausal correlation results are summed, if the ambient-noise sources do not illuminate the receiver stations from all directions with comparable strength or in the stationary-phase regions (Draganov et al., 2013; Wilczynski et al., 2025). Therefore, the summed causal and acausal parts were used to construct the virtual shot gathers of surface waves and body waves.

Figure 5: (a) Noise cross-correlation results with both causal and acausal parts at source location No. 79; (b) Noise cross-coherence results at source location No. 224, DPW and SPW represent the direct or diving P-wave and the direct S-wave, respectively. Numbers mark different quadrants in the time offset coordinate system.

删除[Chris Juhlin]: D 删除[王志辉]: T 删除[Chris Juhlin]: s 删除[Chris Juhlin]: the 删除[Chris Juhlin]: the 删除[evahi56@hotmail.com]: 删除[王志辉]: surface wave and body wave source gathers 删除[王志辉]: (Fig. 5), respectively 删除[evahj56@hotmail.com]: were 删除[evahi56@hotmail.com]: at 删除[evahj56@hotmail.com]: at 删除[evahj56@hotmail.com]: u 删除[Chris Juhlin]: in 删除[Chris Juhlin]: symmetricity 删除[Chris Juhlin]: d 删除[王志辉]: of ambient sources in a frequency band of ... 删除[王志辉]: 5 删除[Chris Juhlin]: s 删除[Chris Juhlin]: retained 删除[王志辉 [21]: . 删除[evahj56@hotmail.com]: and signature

删除[王志辉 [2]]: labeled as

删除[王志辉 [2]]:,

删除[Chris Juhlin]: at

删除[Chris Juhlin]: quarter

删除[Chris Juhlin]: quater

删除[Chris Juhlin]: S

际[Chris Juniin]: S

删除[王志辉 [2]]:

## 4 Surface wave data processing

## 4.1 Dispersion curve extraction

A total of 90 source gathers were retrieved by cross-correlation (Wapenaar and Fokkema, 2006), from receiver number 79 to 358 (see Fig. 1b for locations). Each retrieved source gather consists of 191 nodal units with a 3 s recording length and 0-1900 m offsets. Figure 6a shows a retrieved source gather and its dispersion diagram, the surface wave dominates in the virtual source recording and a reliable frequency curve within the window 0.8-5.5 Hz can be picked (Fig. 6b). The dispersion curves from different source gathers shown in Figure 6c have a good consistency and suggest that we have obtained high signal-to-noise ratio surface wave data and that the structural variation in the horizontal direction is minor in this area.

Figure 6: (a) Retrieved source gather for surface waves at source location No. 79; (b) dispersion diagram from (a); (c) Extracted dispersion curves plotted for all survey points.

#### **4.2 Dispersion curve inversion**

Rayleigh wave inversion is a nonlinear optimization problem. To avoid the solution estimate falling into a local minimum value of the objective function and to minimize the dependency on the initial model, an overall optimization based on a genetic algorithm (Zhao et al., 1995) was used to invert for the shear wave velocity model. Four different models were used as initial models to compare with the inverted results (Fig. 7a). Model 1 used, a velocity gradually increasing with depth, while model 3 was embedded with a high velocity layer at depths of 480 m to 580 m, known to be present from acoustic

删除[王志辉]:

设置格式[王志辉]:字体:(中文)黑体,(中文)中文(简体)

设置格式[王志辉]: 标题1

设置格式[王志辉]:字体:(中文)黑体,(中文)中文(简体)

删除[王志辉]: Calculating and picking of dispersion curves

删除[王志辉]: using equation (1)

删除[王志辉]: 2

删除[王志辉]: 2

删除[王志辉]: 2

删除[王志辉]: 2

删除[王志辉]: D

删除[王志辉]: Inversion for the velocity model

删除[王志辉]: 3

删除[evahj56@hotmail.com]: s

logging data (Fig. 7b and Table 3). Model 2 and model 4 were variations of model 3 and embedded with a high velocity 200

layer between 380 m to 480 m and between 580 m to 680 m, respectively, to test the sensitivity of the inversion to the preferred initial model (model 3). A comparison of the velocity inversion results between model 1 and model 3 shows, as expected, that the inclusion of the high velocity layer in the initial model results in a better match with the acoustic log (Fig. 7b). This is also seen in the comparison between the inverted dispersion curves and the observed one (Fig. 7c and 7e). Furthermore, if this high velocity layer is moved up or down in the initial model then a poorer match between the observed dispersion curves and the inverted curves is obtained (compare Fig. 7e with Fig. 7d and 7f). Consequently, model 3 comprising an eight-layer structure (Table 3) was used as the initial model to invert all dispersion curves to obtain the shear wave velocity image. In the inversion we assume the density of the different sedimentary rocks is constant at 2.7 g/cm<sup>3</sup> and the  $V_p/V_s$  ratio equal is 2 based on well-logging and first break traveltimes. Each extracted frequency dispersion curve was calculated iteratively 30 times to obtain the optimized model. Normalized Root Mean Square Error (NRMSE) described as equation (1) was applied to evaluate the inversion results.

$$\underbrace{NRMSE} = \frac{\sqrt{\sum_{i=1}^{n} (V_{invi} - V_{obsi})^2}}{V_{obs}}$$
(1)

where  $\underline{V}_{invin}$  and  $\underline{V}_{obsin}$  respectively, represent the inverted and observed phase velocity of the i-th frequency point,  $\underline{\overline{V}}_{obsin}$ denotes the mean of all observed phase velocities. As shown in Fig. 7g, NRMSE of all dispersion curves is less than 3% and most of them are around 1%. This demonstrates the robustness of the inversion methodology employed in our study.

删除[王志辉]: 3 删除[王志辉]: starting 删除[王志辉]: 3 删除[王志辉]: 3 删除[王志辉]: 3 删除[王志辉]: 3 删除[王志辉]: 3 删除[王志辉]: 3 删除[王志辉]: Accordingly 删除[evahj56@hotmail.com]: . 删除[Chris Juhlin]: 删除[evahj56@hotmail.com]: . 删除[Chris Juhlin]: We 删除[Chris Juhlin]: that 删除[Chris Juhlin]: kind of 删除[王志辉]: <math> 删除[Chris Juhlin]: constantly 删除[Chris Juhlin]:, 删除[王志辉]: <math> 删除[王志辉]: 删除[Chris Juhlin]: to 删除[Chris Juhlin]: for this experiment 删除[Chris Juhlin]: As shown in Figure. 7g, NRMSE of a ... 设置格式[王志辉]: 右 删除[王志辉]: <math> 删除[王志辉]: <math> 删除[王志辉]: <math>

删除[王志辉]: <math>

Figure 7: (a) Four different initial models for inversion, initial model 1 is a velocity gradually increasing with depth, initial model 2. initial model 3 and initial model 4 are embedded with a high velocity layer at depths of 380 m to 480 m, 480 m to 580 m, and 580 m to 680 m, respectively; (b) Acoustic logging curve and inverted shear wave velocity models Vs1 and Vs3 are from initial model 1 and initial model 3, respectively; (c) Observed and inverted curves from initial model 1; (d) Observed and inverted curves from initial model 2; (e) Observed and inverted curves from initial model 3; (f) Observed and inverted curves from initial model 4; (g) Normalized Root Mean Square Error (NRMSE) for different virtual shot gather inversions after 30 iterations,

删除[王志辉]:

a)

**b**)

设置格式[王志辉]: caption111

设置格式[王志辉]:字体:(中文)宋体,加粗,英语(美 \cdots

删除[王志辉]: 3

删除[王志辉]:

设置格式[王志辉]:字体:加粗,英语(美国)

设置格式[王志辉]:字体:(中文)宋体,加粗,英语(美 …

设置格式[王志辉]:字体:(中文)黑体,加粗,(中文)…

设置格式[王志辉]:字体:(中文)黑体,加粗,(中文)…

设置格式[王志辉]:字体:(中文)黑体,加粗,(中文)…

设置格式[王志辉]:字体:(中文)黑体,加粗,(中文)…

设置格式[王志辉]:字体:(中文)黑体,加粗,(中文)…

| Table 3. Parameters of | f model 3 as initial | model for inversion    | i calculation.  |
|------------------------|----------------------|------------------------|-----------------|
|                        |                      | intouch for interestor | 1 cuicuiutioiii |

| Layer number | $V_{\underline{s}}(m/s)$ | $V_p(m/s)$  | ρ(g/cm <sup>3</sup> ), | Thickness (m)   |
|--------------|--------------------------|-------------|------------------------|-----------------|
| <u>1</u>     | <u>1489</u>              | <u>2978</u> | <u>2.70</u>            | <u>134</u>      |
| <u>2</u>     | <u>1529</u>              | <u>3058</u> | <u>2.70</u>            | <u>80</u>       |
| <u>3</u>     | <u>1686</u>              | <u>3372</u> | <u>2.70</u>            | <u>96</u>       |
| <u>4</u>     | <u>1999</u>              | <u>3998</u> | <u>2.70</u>            | <u>170</u>      |
| <u>5</u>     | <u>3500</u>              | <u>5800</u> | <u>2.70</u>            | <u>100</u>      |
| <u>6</u>     | <u>2521</u>              | <u>5042</u> | <u>2.70</u>            | <u>150</u>      |
| <u>7</u>     | <u>2619</u>              | <u>5238</u> | <u>2.70</u>            | <u>198</u>      |
| Half-space   | <u>2700</u>              | <u>5400</u> | <u>2.70</u>            | <u>Infinite</u> |

## 5 Body wave data processing

A total of 290 source gathers were retrieved with fixed geometry after the cross-coherence calculation. Raw source gather recordings are dominated by low frequency signals (Fig. &a and &e), however, a direct, or diving, P-wave with a velocity of 37,63 m/s, a direct S-wave with a velocity of 1703 m/s, air waves with a velocity of 353 m/s and reflections from the top of Ordovician with a moveout velocity of 3181 m/s are clear in the raw source gather. Reflections with low frequency and high amplitude are present at times of 350 ms and 500 ms. Spectral equalization in the frequency band 5-6-18-19 Hz and a median filter at velocities of 350 m/s were applied on all retrieved raw source gathers to reduce noise and enhance the weaker amplitude reflected signals. Afterward, reflections are more clearly observed in Fig. &c. Further conventional seismic reflection processing included surgical muting, velocity analysis, normal moveout, stacking, f-x deconvolution and bandpass filtering as outlined in Table 2.

For comparison with the passive source gather, an active source gather (Fig. 8b) at the same location is made. Both the active and passive source gathers show the same kinematic characteristics of the seismic waves with similar velocities of the direct P-and S-waves, and reflections from the top of Ordovician. However, a higher velocity of 4200 m/s from the shallow part is shown in the active source gather. This arrival probably represents a wave traveling through the hard near-surface limestone layer (Juhlin et al., 2025). Spherical divergence compensation, deconvolution, bandpass filter, statics, and median filtering were applied to remove noise and enhance reflections in the active source gather (Fig. 8d). Comparison of Fig. 8c and Fig. 8d shows that reflections from the top of Ordovician at about 350 ms correlate well, indicating we are obtaining useful body wave data in the source gathers from the passive data.

/ 删除[王志辉]: <math>

| 删除[王志辉]: <math>

删除[王志辉]: <math>

带格式表格[王志辉]

删除[王志辉]: .

设置格式[王志辉]: 标题 1

删除[王志辉]: 5

设置格式[王志辉]: (中文)中文(简体)

设置格式[王志辉]: (中文)中文(简体)

删除[evahj56@hotmail.com]:

删除[王志辉]: 4

删除[王志辉]: d

删除[王志辉]: 00

删除[王志辉]: 0

删除[王志辉]: and

删除[evahj56@hotmail.com]:

删除[王志辉]: (Fig. 4a)

删除[evahj56@hotmail.com]: 2

删除[王志辉]: as

删除[王志辉]: Direct P-waves and reflections are present ...

删除[王志辉]: now

删除[evahj56@hotmail.com]: ure

删除[王志辉]: .

删除[王志辉]: 4

删除[evahj56@hotmail.com]: C

删除[evahj56@hotmail.com]: flows

删除[Chris Juhlin]: ing

删除[Chris Juhlin]: were carried out

Figure &: Comparisons of active and passive source gathers at shot number, 224. (a) Raw retrieved source gather, DPW and DSW represent the direct or diving P-wave with a velocity of 3763 m/s and the direct S-wave with a velocity of 1703 m/s, reflections from the top of the Ordovician and air waves have velocities of 3763 m/s and 353 m/s, respectively; (b) active source gather with a similar velocity of the direct P-wave and S-wave, reflections from the top of the Ordovician, a higher velocity of 4200 m/s at the shallow part is probably due to hard limestone near the surface; (c) (a) after spherical divergence compensation, deconvolution, bandpass filter, static, median filter to remove noise and enhance reflections; (d) (b) after spectral equalization filter (5-6-18-19 Hz) and median filtering to remove noise, air wave at 350 m/s; (e), (f), (g), (h) are power spectra for (a), (b), (c) and (d), respectively.

## 6 Synthetic seismogram

The acoustic velocity log from the Nore-1 borehole along with a constant density value,  $2.7 \times 10^3$  kg/m³, was used to calculate a seismic impedance log and associated reflection coefficients (Fig. 9a). A synthetic seismogram was then produced by convolving the acoustic reflection coefficient with a 10 Hz Ricker wavelet (Fig. 9b). To compare with the passive seismic results, 11 stacked CDP gathers from near the Nore-1 borehole were extracted from the passive stacked section (Fig. 9c). As shown in Fig. 9b and Fig. 9c, three reflections correlate well between the synthetic seismogram and the passive seismic image at times of 170 ms, 280 ms and 360 ms, indicating that the passive seismic reflection data are trustworthy and can be used for geological interpretation.

删除[王志辉]: 4 删除[Chris Juhlin]: the 删除[Chris Juhlin]: of 删除[王志辉]: Noise attenuation processing steps from \cdots 删除[王志辉]: 00 删除[王志辉]: 0 删除[Chris Juhlin]: the 删除[Chris Juhlin]: the 删除[Chris Juhlin]: 删除[evahj56@hotmail.com]: maybe comes from 删除[Chris Juhlin]: frozen soil 删除[evahj56@hotmail.com]: a 删除[Chris Juhlin]: B 删除[Chris Juhlin]: 删除[Chris Juhlin]: s 删除[Chris Juhlin]: make 删除[evahj56@hotmail.com]: dominated 删除[王志辉]: spectral equalization filter (5-6-18-19 H ...) 删除[Chris Juhlin]: 4 删除[王志辉]:, 删除[王志辉]: and 删除[evahj56@hotmail.com]: um 删除[王志辉]: a 删除[王志辉]: 删除[王志辉]: and 删除[王志辉]: 5 删除[王志辉]: 5

Figure 2: (a) Reflection coefficient curve based on acoustic velocity logging from the Nore-1 borehole and an assumed constant density of 2.7 ×10<sup>3</sup> kg/m<sup>3</sup>. (b) Synthetic seismogram produced from convolving (a) with a 10 Hz Ricker wavelet. (c) stacked CDP gathers near borehole Nore-1.

删除[王志辉]: 5

#### 7 Results

# 7.1 Shear wave velocity model

The extracted surface wave data were inverted for shear wave velocity using the logging data as constraints. The inverted shear wave velocity model shown in Fig. 10a can be interpreted as representing four geological units that can be correlated well with the borehole section (Fig. 10b). From the ground surface to 470 m depth, the velocity increases and varies from 1400 m/s to c. 2600 m/s. A high velocity layer is embedded at the depths of 470 m to 550 m, which corresponds to the Ordovician. Note that the presence of this high velocity layer, and the depth to it, is dependent upon the starting model used, but the results indicate that it is continuous along the profile and the depth to it does not vary. Cambrian rocks are located from 550 m to 780 m. The high velocity under 780 m represents the Precambrian. Moreover, the velocity structure is continuous along the survey line, indicating that there are no potentially large-scale faults in the area. The reliable geological interpretation suggests that passive surface wave inversion results can be used to map the geological strata in the area. Although not proved here, they also have the potential to map regional faults.

删除[王志辉]: Inverted shear wave velocity model

删除[evahj56@hotmail.com]: ure

删除[王志辉]: .

删除[王志辉]: 6

Figure 10: (a) Inverted shear wave velocity from passive surface waves. (b) Nore-1 borehole section.

#### 7.2 Stacked seismic reflection section

Passive source gathers containing body waves, were processed to test if the passive body wave image can resolve subsurface geological structure. Acoustic logging analyses from Nore-1 at depths of 170 m to 800 m indicated three seismic reflections at travel times of c. 170 ms, 280 ms and 360 ms (labelled as R1, R2 and R3 in Fig. 11) can also be mapped in the stacked section. The uppermost two reflections (R1 and R2) are produced by marlstone with interbeds of wackestone and mudstone from the Silurian. The reflection labelled as R3 likely originates from the top of the Ordovician, consistent with logging data (Fig. 9b) and active data (Fig. 12). In addition to the R1, R2 and R3 reflections, two more reflecting horizons are recognized at times of c. 500 ms and 700 ms, labelled as R4 and R5 in Fig. 11. R4 may represent the base of the Cambrian that was interpreted in the active data as the point where the seismic response in the stacked section becomes generally transparent (Juhlin et al., 2025). No borehole or other data are available to validate where R5 originates from, it may originate within the Precambrian or represent a multiple from the top of the Ordovician. As for the inverted shear wave velocity model (Fig. 10a), all reflection horizons shown in Fig. 11 are nearly flat with no obvious disturbances, indicating no large-scale faults are present again.

删除[王志辉]: 6
删除[王志辉]: (Fig. 4)
删除[王志辉]: 5
删除[王志辉]: 5
删除[王志辉]: , as shown in Figure. 711
删除[王志辉]: labelled
删除[王志辉]: 8
删除[Chris Juhlin]: another
删除[evahj56@hotmail.com]: ure
删除[王志辉]: .
删除[王志辉]: 7
删除[王志辉]: 6
删除[evahj56@hotmail.com]: ure

删除[王志辉]: .

Figure 11: Passive seismic reflection section overlaid with its power spectrum. R1 and R2 represent reflections from the interior of Silurian, R3 represents a reflection from the top of the Ordovician, R4 represents a reflection from the base of the Cambrian, and R5 represents a possible reflection from within the Precambrian or a multiple from the top of the Ordovician.

Figure 12: Active seismic reflection stacked section overlaid with its power spectrum (after Juhlin et al., 2025). R1 and R2 represent reflections from within the Silurian, R3 represents a reflection from the top of the Ordovician, R4 represents a reflection from the base of the Cambrian, and R5 represents a possible reflection from within the Precambrian or a multiple from the top of the Ordovician. The dashed box indicates reflected energy that can potentially be correlated with the R5 reflection in the passive stacked section in Figure 11.

删除[王志辉]: 7

删除[王志辉]:

设置格式[王志辉]: caption111

删除[王志辉]: 8

设置格式[王志辉]:字体:(中文)宋体,加粗,英语(美 …

设置格式[王志辉]:字体:(中文)黑体,加粗,(中文)…

删除[Chris Juhlin]: the interior of

设置格式[王志辉]:字体:(中文)宋体,加粗,英语(美 …

设置格式[王志辉]:字体:(中文)宋体,加粗,英语(美 ...

## 8 Discussion

# 8.1 Comparisons of active and passive seismic reflection imaging

The stacked image from the high-resolution active seismic shown in Fig. 12 (Juhlin et al., 2025) indicates that a particularly strong reflection at about 350 ms likely originates from the top of the Ordovician. In addition, Cambrian sandstones below are also reflective, as well as shallow sandstone layers in the upper 150 ms. Reflections labelled R1, R2 and R3 in the active data are consistent with the passive seismic stacked section (Fig. 11). As mentioned previously, R4 may represent a reflection from the top of the crystalline basement. No clear top of basement reflection is observed in the active data, but this may be due to the lack of lower frequencies in these data or the power limitation of the weight drop hammer. The lower frequencies in the passive data may be more sensitive to the velocity structure near the top of the crystalline basement. There is no clear reflection in the active seismic data that corresponds to the R5 reflection in the passive data. Some indications of reflected energy, interpreted to lie below the crystalline basement can be observed in the active data within the dashed rectangle in Fig. 12. However, it should be noted that the timing of this reflected energy is consistent with a multiple from the top of the Ordovician. The general consistency of the passive seismic data with the active data down to the top of the crystalline basement implies that passive methods can be used to reliably map some of the structural features on Gotland. However, the active data provide a higher resolution image and more geological detail. The power spectra of the passive and active reflection data shown in Fig. 12 indicate the dominant frequency ranges for the passive data are 5-20 Hz, while the active data have a 30-200 Hz range. Consequently, we cannot expect that the passive seismic reflection method will provide the same resolution as the active seismic reflection method, but the methodology can be used to map the general structure of the Palaeozoic rocks at a low cost.

## 8.2 Relevance of results for CO<sub>2</sub> storage and seismic imaging

Both the passive surface wave and body wave results provide high-quality images for investigating the subsurface geological structure at the Sudret site. Processing of the passive data allowed us to obtain images of the subsurface down to the Precambrian basement around the Nore-1 borehole that had been core-drilled earlier down to about 800 m. Compared with the active data (Juhlin et al., 2025), these images show major features that can be mapped over a wide area at low cost using passive methods. However, the Sudret site may not represent a typical site for passive seismic imaging given that it is located on an island. This allows noise from nearly all directions to be recorded, which is especially important for body wave imaging. Furthermore, the ambient noise from local sources was very low at the location, allowing far-field noise to be recorded without interference. A next step would be to set up synthetic models to test how much CO<sub>2</sub> is required in the bedrock in order to detect it by the passive methods presented in this paper. The surface waves would be more sensitive to pressure changes in the reservoir, whereas the body waves would be more sensitive to the amount of free CO<sub>2</sub> in the reservoir. It remains to be seen if these passive methods can be used for monitoring CO<sub>2</sub> storage sites in an effective manner.

删除[王志辉]: 8.1 Analysis of frequency and direction of \cdots 删除[王志辉]: 2 删除[evahj56@hotmail.com]: ure 删除[王志辉]: . 删除[王志辉]: 8 删除[王志辉]: 1 删除[王志辉]: 7 删除[王志辉]: 0 删除[Chris Juhlin]: shortage 删除[王志辉]: the higher frequency nature 删除[Chris Juhlin]: of 删除[evahj56@hotmail.com]: ure 删除[王志辉]:. 删除[王志辉]: 8 删除[王志辉]: 1 删除[evahj56@hotmail.com]: ure 删除[王志辉]: . 删除[王志辉]: 7 删除[王志辉]: 0 删除[evahj56@hotmail.com]: ure 删除[王志辉]: . 删除[王志辉]: 8 删除[王志辉]: 1 删除[王志辉]: 3 删除[evahj56@hotmail.com]: in the vicinity of 删除[evahj56@hotmail.com]: at 删除[王志辉]: the southern tip of

删除[evahi56@hotmail.com]: for it to be detectable

## 335 9 Conclusions

We retrieved virtual shot gathers with body waves and surface waves after applying signal separation through cross-coherence and cross-correlation calculations. For the body waves, conventional seismic data processing was conducted to obtain a stacked section consistent with active data and a synthetic seismogram generated from an acoustic sonic log. For the surface waves, we determined the dispersion curve in the frequency range 0.8 to 5.5 Hz and inverted these curves to obtain a velocity model that correlates with borehole data from the surface down to c. 800 m. Both the body waves and surface waves provide high-quality images of the top of the Ordovician formation and have a good consistency with the borehole section. These results show that passive data can be used for mapping some general features in the subsurface of Gotland. Specifically, the top of the Ordovician and the top of the Precambrian can be mapped. Compared with active seismic exploration, passive seismic is friendly to the environment and cost effective. In some cases, it may have the potential to replace active seismic imaging for initial subsurface surveying of the sedimentary layers on Gotland, and perhaps elsewhere. The method remains to be tested for monitoring CO<sub>2</sub> storage sites.

Data availability

Requests for the seismic data should be directed to SGU (www.sgu.se)

#### **Author contribution**

CJ and PH conceptualized and designed this study. ZW and CJ were involved in the data acquisition and responsible for the data processing. ME and DS provided borehole data. ZW and CJ led the geological interpretation. ZW wrote the initial draft and CJ reviewed it. All authors participated in the results discussion and approved the submission of this paper.

#### **Competing interests**

The authors acknowledge that there are no conflicts of interest.

## 355 Acknowledgments

This research was funded by the Deep Earth Probe and Mineral Resources Exploration - National Science and Technology Major Project (2024ZD1002201), the Geological Survey of Sweden and the China Geological Survey Project (DD20221819). We acknowledge Dr. Wilczynski Zbigniew from Uppsala University, Dr. Bojan Brodic from University of the Witwatersrand, South Africa and Johan Söderman and Per Wahlquist from the Geological Survey of Sweden who contributed to the data acquisition. Globe Claritas<sup>TM</sup> under the academic license from Petrosys Ltd. and Seismic Unix was

删除[evahj56@hotmail.com]:

删除[evahj56@hotmail.com]: In particular

删除[evahj56@hotmail.com]:

删除[王志辉]:

删除[王志辉]: the Deep Earth Probe and Mineral Resources Exploration - National Science and Technology Major Project (2024ZD1002201)

删除[Chris Juhlin]:

used for the data processing. We thank two anonymous reviewers whose constructive comments helped to improve this paper.

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
