# Peer review of "Passive seismic imaging of the Lower Palaeozoic in the Sudret area of Gotland, Sweden"

_EGUsphere, 2025_

## Author Response (AR1)

**Dear Editors and Reviewers:**

Thanks for your comments and professional suggestions concerning the manuscript entitled "Passive seismic imaging of the Lower Palaeozoic in the Sudret area of Gotland, Sweden". Those comments and suggestions are all valuable for us to revise and improve our paper, as well as important for improving the readability of the paper. We have made corrections and improvements according to the comments and provide our responses point by point as follows.

**Responses to referee comments #1**

**Comment 1:** Figure 1. Add labels for Figure 1(b) (e.g., x (m) and y(m)). If possible, a lighter background color will make the symbols more straightforward to distinguish from the background.

Response 1: We have revised Figure 1 following your suggestion (see page 4 line 88).

**Comment 2:** Line 90. Since the operating period is 14 hours, it is better to state the exact start time and end time of the data.

**Response 2:** Thanks. We have stated the exact time form 17:00 on 12 November, 2023 to 07:00 on 13 November, 2023 (UTC+1) instead of 14 hours (see page 4 line 100).

**Comment 3:** Line 122. Describe the strengths and weaknesses of two methodologies and why two approaches were conducted for the surface and body wave extraction, with appropriate references.

**Response 3:** We cited some references to describe the strengths and weaknesses of the two methodologies to explain why we retrieved surface waves through cross-correlation calculation and body waves through cross-coherence (see page 9 lines 162 - 165).

**Comment 4:** Line 208. The figure citation here may be Figure 7, not Figure 5. I wonder why the numbers 150, 280, and 360 ms here do not match 170, 280, and 360 ms in line 185 for the synthetic seismogram section.

Response 4: Thanks for pointing out the mistakes. We have corrected them (see page 16 line 280).

**Comment 5:** Figures 7 and 8: To contrast the background seismic images, it would be good to use a color other than green for the fonts.

**Response 5:** Thanks again. We changed drill line color into white and removed the fonts in two figures and gave the description for the reflectivity of R1, R2, R3 and R4 (see page 17).

**Comment 6:** Line 236: It would be good to have references for the relevance of 1, 5, 12, and 16 Hz originating from ocean waves and human activities. Can we assign the four frequencies to either

ocean waves or human activities?

**Response 6:** Unlike the active seismic data with certain energy and location of sources, passive data rely on natural and anthropogenic sources, such as seismic energy generated from ocean-wave-induced microseisms (0.03 - 1 Hz), earthquakes (< 1 Hz), traffic (> 1 Hz) and so on (Nimiya et al., 2021; Bertoldi et al., 2024). The research area is located on an island and surrounded by the Baltic sea, so the noise source with the frequency of 0.7-1 Hz is probably ocean waves (see page 5 lines 112 - 115).

**Comment 7:** Line 278: Why does the geometric location of sensors at the southern tip of an island allow noise from nearly all directions?.

**Response 7:** This is because the ambient noise sources are from different countries around the island of Gotland, such as mainland Sweden, Finland, Estonia, Poland, Germany, Denmark, Norway, etc (see page 4 line 88, Fig. 1a).

**Responses to referee comments #2**

**General comments**

**Comment 1:** The authors don't justify why they separated data processing into the evaluation of surface wave and body wave parts of empirical Green's functions (EGF). What is the reason to use different formulas to evaluate these parts? They should justify this by the results of previous research or their observations. Why is cross-correlation better for extracting surface waves but cross-coherence for body waves? Is it possible to evaluate the surface wave part of the empirical Green's function by cross-coherence and the body wave part by cross-correlation? Why is it impossible to have the same procedure for both types of waves?

Response 1: Thanks for your questions. A lot of comparisons have been conducted by many researchers, such as Snieder et al.(2009), Nakata et al.(2011), Zhang et al.(2019). In their research, they describe the strengths and weaknesses of cross-correlation and cross-coherence and state that cross-correlation is stable, clearer and less noisy and cross-coherence can enhance temporal resolution and be able to retrieve the body wave data. So we followed their research results and retrieved surface wave through cross-correlation and body wave through cross-coherence. Some appropriate references were cited in the revised manuscript (see page 9 line 164). Also, surface wave and body wave parts of NCFs with both causal and acausal parts are now provided (see page 9 line 178, Fig. 5).

**Comment 2:** They don't prove that they extracted reflected waves. I am concerned that the ambient noise, analysed by them, consists of only surface waves, because the dominant source of seismic energy is the ocean in their case. Despite mentioning anthropogenic sources, they don't provide any polarisation analysis of ambient noise and EGF. In my opinion is more reasonable to split the whole record into "body wave dominated" windows of ambient noise, and "surface wave dominated" windows. This is possible to do with polarisation analysis of ambient noise seismic records in the selected frequency band. After this is possible to cross-correlate them separately and stack them

accordingly to separate the body wave and surface wave parts of EGF. In my opinion, separating them by applying cross-correlation and cross-coherence is not possible.

Response 2: Thanks again. It is much more difficult to retrieve body waves than surface waves from ambient noise data, but extracting body waves from ambient noise data is possible in some cases. However, we are lucky to be recording not only surface wave but also body waves (see page 6 line 120, Fig. 2) included in our ambient noise data. Their EGFs (see page 9 line 175, Fig. 5) are also provided to validate the virtual shot gathers. Presently, we cannot conduct polarisation analysis of the ambient noise since only 1C sensors were used to record the vertical component. We provide the synthetic seismogram and active stacked section to support the claim that we extracted body waves from the ambient noise data. Comparisons of active and passive shot gathers are now presented in the revised manuscript (see page 14 line 243, Fig. 8).

**Comment 3:** Analysis of directivity and frequency content of ambient noise should be done prior to cross-correlation. Providing this analysis in the discussion part makes reading confusing. Difficult to understand why they selected particular frequency ranges and the EGF evaluation method. I recommend moving this text from the discussion part to a separate chapter (after the description of the experiment).

**Response 3:** We totally agree with your comment 3 that the directional and frequency content of the ambient noise data have be moved prior to cross-correlation and cross-coherence (see page 5 line 105 to page 7 line 140).

**Comment 4:** The authors calculated cross-correlation functions in the frequency domain, which is wrong in their case. This is possible only for the case of symmetric EGF, i.e. in the case when sources of ambient seismic noise are distributed almost homogeneously on azimuths. This is justified in experiments where ambient noise records have the length of months or years, but not for short-term experiments. From the azimuthal distribution presented by the authors, it's obvious that the sources are mainly outside of the Fresnel zone. In that case, only cross-correlation in the time domain with further velocity correction can be applied.

**Response 4:** Thanks for your comments, but we respectfully disagree with some of your points. As shown in Figure 4, sources of ambient noise contributing to extracting surface waves with a velocity less than 4000 m/s came from the NNW and the azimuth of the ambient noise sources are consistent with the seismic survey line and located in the stationary phase zone, so we can calculate cross-correlation functions in the frequency domain and do not need to correct for velocity.

**Comment 5:** Authors don't show signal (EGF). Therefore is difficult to estimate the correctness of dispersion curves. Moreover, they don't provide enough information about dispersion curve inversion, which makes it difficult to estimate the quality of the results.

Response 5: Thanks for your question. We have provided an EGF (see page 9 line 178) and much more information about the data processing and inversion to validate our results (see page 11 lines 205 - 210 and page 12 line 220).

Comment 6: In the conclusion part authors write "A next step would be to set up synthetic models to test how much  $CO_2$  is required in the bedrock in order for it to be detectable by the passive methods presented in this paper". By this conclusion, they state that they did not obtain the expected results. What is the reason to publish this manuscript in that case? I recommend conducting a significant study with synthetic data first. For example, they could calculate synthetic seismograms of waves propagating through the synthetic model of the considered environment (fortunately, they have drillhole data) with different content of  $CO_2$ . These results will help in understanding limitations, and authors should include these results in the manuscript. In that case would be possible to make clearer conclusions about the structure of the studied environment.

**Response 6:** Thanks. This paper focuses on presenting passive seismic imaging of the Lower Palaeozoic in the Sudret area, and provides a friendly to the environment and cost effective complement for active seismic. In the discussion part, we discuss the future work to monitor  $CO_2$  injection. However, synthetic modeling is beyond the scope of this paper currently. And surface waves are not very sensitive to the presence of  $CO_2$  since replacement of saline water by  $CO_2$  does not change the S-wave velocity very much.

**Specific comments**

**Comment 1:** P1 L14-16. The sentence should be rephrased. Seismic waves cannot provide geological information. Their interpretation can provide structural information about the geological medium. What does "geological information complement active data" mean? What is low low-frequency image? Maybe "image, obtained by analysing the signal of low frequency"? "The passive data are consistent with active data" Better to rephrase: "Results obtained by analysing controlled source seismic data and passive seismic data are in a good correlation".

Response 1: Thanks for your comments and good suggestion. We rephrased these sentences (see page 1 lines 14 - 18).

**Comment 2:** P1 L20 "evaluating" sounds strange. "for structural studies of near surface " sounds better.

Response 2: Thanks. We have corrected it following your suggestion (see page 1 lines 21).

**Comment 3:** P2 L55 "limited source energy" This is a poorly justified statement. It is possible to use an artificial source of high enough energy to study the crust down to dozens of kilometres.

**Response 3:** Thanks. We meant that our source, a 500 kg weight drop hammer, has limited source energy and penetration depth (see page 2 line 59).

Comment 4: P3 L69 "Ordovician generates are clear reflection". Structure itself cannot generate

reflected waves, but it can reflect waves. Better to write: "The top of the Ordovician has strong reflectivity according to results of controlled source seismic data analysis".

Response 4: Thanks. We have changed the wording to better describe the physics (see page 3 line 73).

**Comment 5:** P3 L71-72 The sentence is unclear. Please rephrase.

Response 5: We have rephrased the sentence (see page 3 lines 75 - 77).

**Comment 6:** P5 Table 1. What do the shot interval and the number of source points in passive seismic data mean?

Response 6: It means the spacing interval and number of retrieved shots (see page 5 line 102, Table 1).

**Comment 7:** P 7 "4.1. Calculating and picking of dispersion curves" better to write "Dispersion curve extraction"

Response 7: Thanks. "Dispersion curve extraction" is much better (see page 10 line 183).

Comment 8: P 8 "4.2. Inversion for the velocity model" better to write "Dispersion curve inversion".

Response 8: Thanks. We rewrote it with "Dispersion curve inversion" (see page 10 line 193).

**Comment 9:** P 10 Figure 4. Distances instead of channels are more useful. Arrivals of all phases and apparent velocities should be added to the plot.

Response 9: Figure 4 has been revised (see page 14 line 243, Fig. 8).

Comment 10: P10 Figure 4 caption. "noise reduction" sounds better than "noise attenuation"

**Response 10:** Yes, we revised Figure 4 and its caption (see page 14 lines 244 - 250).

Comment 11: P 11 "7.1. Shear wave velocity model" sounds better.

Response 11: Thanks. We have corrected it (see page 15 line 264).

**Comment 12:** P 11 What initial model (parameters of the medium) did you use for dispersion curve inversion? Better to provide this as a table. More details are needed about the inversion process.

Response 12: The parameters and methods of data processing and inversion have been added to the

manuscript (see pages 11 - 13).

**Comment 13:** P13 There is poor correlation between sections obtained by analysing passive and controlled source seismic data. Only one boundary is visible on both sections. I guess this is because of the absence of reflected waves on the ambient noise. The authors did not prove that they extracted body waves. I would recommend conducting polarisation analysis of the extracted signals.

**Response 13:** We cannot expect that the passive seismic reflection section provides the same result as active data. In the revised manuscript, we removed fonts in the Figure 11 and Figure 12 to make it much clearer (see pages 17). They have good correlations at R1, R2, R3 and R4. Currently, we cannot provide polarisation analysis of the extracted signals because only 1C sensors were used to collect data in our study.

**Comment 14:** P14 L236-237 "Particularly, body waves and surface waves can be separated from each other at a frequency of 7 Hz." How do you separate these waves?

**Response 14:** Thanks. Surface waves usually dominate in ambient noise data, so we just apply a bandpass filter with corner frequencies of 0.5-1-40-50 Hz to filter out too low/high-frequency components to avoid producing noise during cross-correlation calculation and followed before cross-correlation calculation. For body waves, we used a bandpass filter of 5-6-20-25 Hz to remove surface waves and extract body waves based on their spectra (see page 6 line 124, Fig.3 and page 7 lines 130).

**Comment 15:** P15 L246. "maximum values of the beam forming analysis in...". Do you mean "maximum beam power"?

Response 15: Yes, it means maximum beam power (see page 7 line 130).

**Comment 16:** P 15 Figure 10. The sources' azimuths are mainly near-perpendicular to the profile in subplot b. Why are such differences in velocities observed? Have you corrected the phase velocities, taking into account this distribution? This also shows that formula (1) cannot be used to evaluate empirical Green's functions.

**Response 16:** For surface wave frequency, the sources' azimuth (Figure 4a) is mainly along NNW, consistent with the seismic survey line and located in the stationary phase zone (see page 7 line 137, Fig. 4a). So we did not correct the phase velocities.

**Comment 17:** P 15 L260-261. "No clear top of basement reflection is observed in the active data, but this may be due to the higher frequency nature of these data." Why? How does reflectivity correlate with the frequency of the signal?

Response 17: Reflectivity does not correlate with the frequency of the signal. We intended to state

that "No clear top of basement reflection is observed in the active data, but this may be due to the lack of lower frequencies of these data and the power limitation of the weight drop hammer". If the transition to the basement is gradual then lower frequency signals may reflect whereas higher frequency ones will not (see page 18 lines 310 - 312).

Thank you very much for your attention and time. Do not hesitate to contact us if you have any questions about the revised manuscript. Looking forward to hearing from you.

Best Regards,

Zhihui Wang